# Model-based prioritization for acquiring protection

**Sarah M. Tashjian** [1]*, **Toby Wise**[1,2], **Dean Mobbs**[1,3]

**1** Humanities and Social Sciences, California Institute of Technology, Pasadena, California, United States of America, **2** Department of Neuroimaging, Institute of Psychiatry, Psychology and Neuroscience, King's College London, London, United Kingdom, **3** Computation and Neural Systems, California Institute of Technology, Pasadena, California, United States of America

* smtashji@caltech.edu

## Abstract

Protection often involves the capacity to prospectively plan the actions needed to mitigate harm. The computational architecture of decisions involving protection remains unclear, as well as whether these decisions differ from other beneficial prospective actions such as reward acquisition. Here we compare protection acquisition to reward acquisition and punishment avoidance to examine overlapping and distinct features across the three action types. Protection acquisition is positively valenced similar to reward. For both protection and reward, the more the actor gains, the more benefit. However, reward and protection occur in different contexts, with protection existing in aversive contexts. Punishment avoidance also occurs in aversive contexts, but differs from protection because punishment is negatively valenced and motivates avoidance. Across three independent studies (Total $N = 600$) we applied computational modeling to examine model-based reinforcement learning for protection, reward, and punishment in humans. Decisions motivated by acquiring protection evoked a higher degree of model-based control than acquiring reward or avoiding punishment, with no significant differences in learning rate. The context-valence asymmetry characteristic of protection increased deployment of flexible decision strategies, suggesting model-based control depends on the context in which outcomes are encountered as well as the valence of the outcome.

## Author summary

Acquiring protection is a ubiquitous way humans achieve safety. Humans make future-oriented decisions to acquire safety when they anticipate the possibility of future danger. These prospective safety decisions likely engage model-based control systems, which facilitate goal-oriented decision making by creating a mental map of the external environment. Inability to effectively use model-based control may reveal new insights into how safety decisions go awry in psychopathology. However, computational decision frameworks that can identify contributions of model-based control have yet to be applied to safety. Clinical science instead dominates and investigates decisions to seek out safety as a maladaptive response to threat. Focusing on maladaptive safety prevents a full

**Data Availability Statement:** Task and model code as well as raw data are available through OSF, https://osf.io/4j3qz/.

**Funding:** DM and SMT are supported by the US National Institute of Mental Health grant no. 2P50MH094258 and Templeton Foundation grant

TWCF0366. TW is supported by a Professor Anthony Mellows Fellowship. The funders had no role in study design, data collection and analysis, decision to publish, or preparation of the manuscript.

**Competing interests:** The authors have declared that no competing interests exist.

understanding of how humans make decisions motivated by adaptive goals. The current studies apply computational models of decision control systems to understand how humans make adaptive decisions to acquire protection compared with acquiring reward and avoiding threat. Safety-motivated decisions elicited increased model-based control compared to reward- or threat-motivated decisions. These findings demonstrate that safety is not simply reward seeking or threat avoidance in a different form, but instead safety elicits distinct contributions of decision control systems important for goal-directed behavior.

## Introduction

Humans have a remarkable capacity to foresee and avoid harm through protective strategies [1]. Acquiring protection through prospective decisions is a predominant way humans achieve safety: we build fences to keep out dangerous animals, buy weapons to defend against conspecifics, and wear protective clothing to shield from natural elements. These protective behaviors likely utilize model-based decision control systems, which support goal-directed action. However, to our knowledge, no prior work has examined how computational control systems support adaptive protection acquisition. Extant literature on decision making focuses on purely appetitive or aversive domains such as reward acquisition and punishment avoidance. When considering safety, the focus is either on safety as threat cessation or on maladaptive safety decisions. These approaches fail to consider how humans adaptively acquire safety, which is a ubiquitous decision process important for survival. We address this gap by comparing protection acquisition with reward acquisition and punishment avoidance using a novel set of tasks designed to test contributions of model-based and model-free control systems to decision making.

Computational reinforcement learning frameworks are powerful methods for characterizing decision making that have yet to be applied to protection. Existing literature provides strong evidence that reward- and punishment-motivated decisions are underpinned by model-based and model-free control systems [2,3]. With model-free control, positively reinforced actions are repeated when similar stimuli are subsequently presented. The resulting habit-like actions are stimulus-triggered responses based on accumulated trial and error learning rather than goal-directed responses. The model-based system, in contrast, builds a map of the environment and uses that map to prospectively determine the best course of action. The model-free system is computationally efficient whereas the model-based system is computationally intensive but highly flexible [4]. Individuals tend to use a mix of strategies with differences dependent on task demands [5], development [6], and psychiatric symptomology [7].

Identifying the extent to which individuals engage in model-based control for protection acquisition is an important step in understanding how humans make adaptive safety-related decisions. Effectively mapping protection contingencies through model-based control is posited to create a positive feedback loop: prospectively acquiring protection extends the capacity for evaluating and integrating knowledge about the environment when later circumstances limit time to engage the model-based system [8]. Over-reliance on model-free systems, by contrast, may underlie pathological as opposed to adaptive protection-seeking, which is characterized by repeating safety behaviors that are disproportionate to the threat faced [9]. Learned helplessness can also manifest as model-free prioritization in cases where increased model-based control does not result in more advantageous outcomes [10]. An unresolved question centers on whether of the unique features of protection shift the degree to which the model-based system or the model-free control system dominate.

Protective decisions are distinct but retain superficial similarities to both reward- and punishment-motivated decisions. Protection is positively-valenced, similar to reward but unlike punishment. Protection exists in a negative context, similar to punishment but unlike reward (Fig 1A). Additionally, protection is distinct in the degree to which valence and behavior are aligned, which has consequences for learning [11–13], Negatively-valenced stimuli like punishment typically elicit avoidance behaviors, whereas positively-valenced stimuli like reward elicit approach behaviors [14]. In this study, subjects were incentivized to actively seek out the maximum protection available as opposed to avoid the highest punishment. This aligns with traditional definitions of approach motivation as the energization of behavior toward a positive stimulus [15]. Prior studies of decision control typically exploit the conventional coupling of valence and context (i.e., positively-valenced outcomes in an appetitive context or

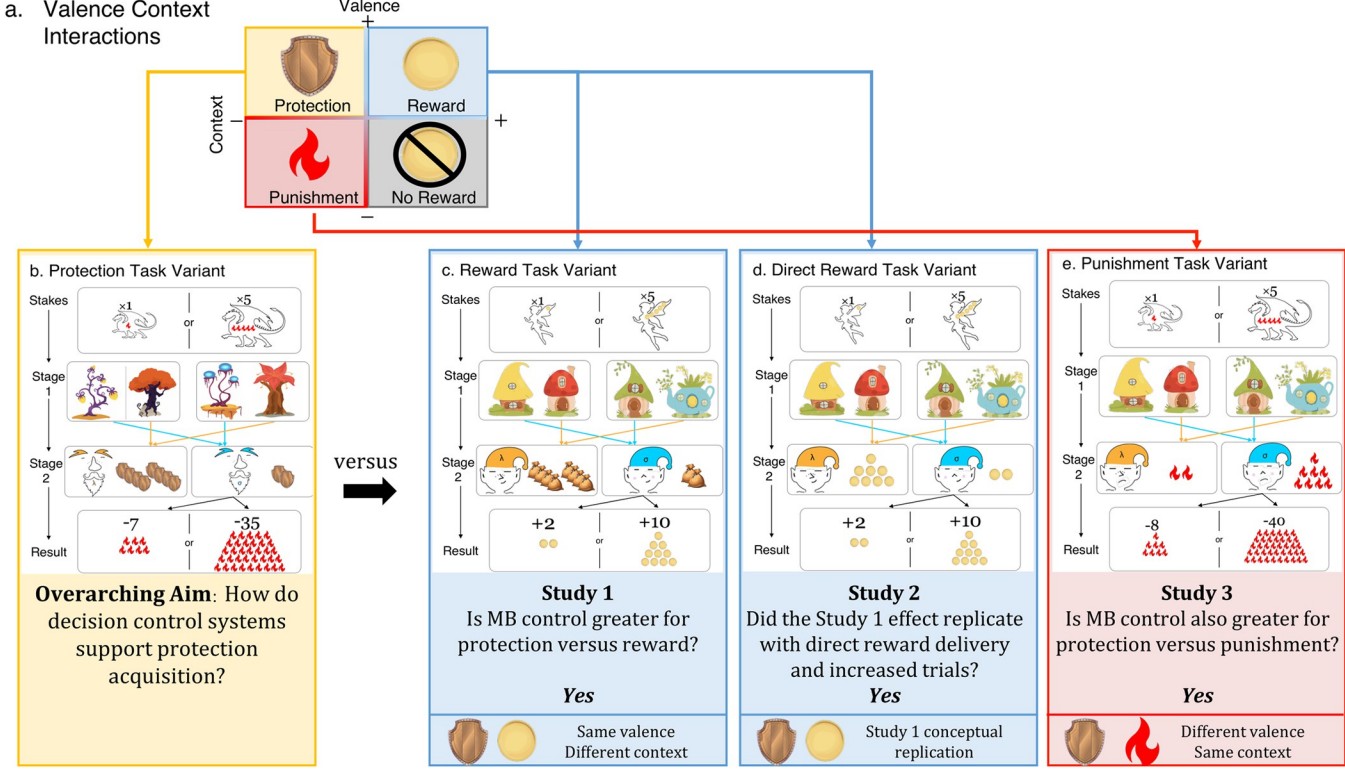

**Fig 1. Study structure. (a)** Protection acquisition shares positive valence features with appetitive reward and negative context features with aversive punishment. The context-valence asymmetry of protection acquisition was hypothesized to be reflected in distinct engagement of decision control systems compared with stimuli in consistently appetitive or aversive domains. All studies included the **(b)** protection acquisition task variant and a comparison task variant: **(c)** reward acquisition in Study 1, **(d)** direct reward acquisition in Study 2, and **(e)** punishment avoidance in Study 3. Study 1 compared protection and reward (b versus c) using abbreviated task versions comprised of 100 non-practice trials. Study 2 compared protection and direct reward (b versus d) using longer task versions comprised of 200 non-practice trials. Study 3 compared protection and punishment (b versus e) using the longer task versions comprised of 200 non-practice trials. Deterministic transition structures are depicted with blue and orange arrows and indicate that the same first-stage state always leads to the same second-stage state. At the start of each trial, subjects saw the stakes amplifier, which showed "x1" for low-stake trials or "x5" for high-stake trials. Low-stakes results ranged from 0–9 units whereas high-stakes results ranged from 0–45 units. The stakes amplifier was applied to the punishment/reward available on that trial as well as the final result received. Next, subjects saw one of two pairs of first-stage dwellings (e.g., trees or houses). After subjects chose between the left and right dwelling depicted, they transitioned to the second-stage creature (e.g., gnomes or elves). Second-stage creatures delivered outcomes in the form of shields (protection), sacks (reward), coins (direct reward), or flames (punishment). At the second-stage, subjects received outcomes ranging between 0–9 according to a drifting outcome rate. Outcomes changed slowly over the course of the task according to independent Gaussian random walks ($\sigma =$ 2) with reflecting bounds at 0 and 9 to encourage learning throughout. Outcomes were multiplied by stakes and presented as final results applied to the maximum reward/penalty available on each trial. For example, in panel (b), subjects visited the low-payoff second-stage gnome. This gnome delivered two shields. When two shields were delivered on a low-stakes trial, which had the threat of 9 dragon flames, the end result was 7 flames (9 minus 2). When two shields were delivered on a high-stakes trial, which had a threat of 45 dragon flames (9 flames multiplied by the stakes amplifier of 5), the end result was 35 flames (45 minus 10, 10 is calculated from 2 shields multiplied by the stakes amplifier of 5).

negatively-valenced outcomes in an aversive context). This perspective does not sufficiently identify how decision control systems contribute to acquiring protection because protection decisions involve asymmetric valence and context.

Protective decisions are also largely absent from traditional conceptualizations of safety, which consider the cessation of punishment but do not consider circumstances in which punishment is reduced through the conferral of positive protective stimuli [1]. Thus, it remains an open question whether the decision control systems for protection differ from reward, with which it shares a positive valence, or from punishment, with which it shares a negative context. It is possible that the context-valence asymmetry of protection has no effect on the computational decision structure when compared with these traditional stimuli. In prior work, reward acquisition and punishment avoidance elicit similar weighting of model-based control, suggesting that there may be some common substrate for reinforcement learning irrespective of stimulus properties [11,16]. However, reward and punishment are valence-context congruent. This valence-context symmetry[17] could favor model-free control as a result of less complex contingency learning [5,13]. Support for this hypothesis is evident in predictable environments where reward learning engages goal-directed control early on, but cedes to habitual control as an efficiency [18]. By contrast, amplification of prospective model-based control can aid development of accurate and flexible action policies [11], which may facilitate response to hierarchically-organized motivational demands (approach toward protection with a broader goal to avoid punishment). Thus, the valence-context asymmetry of protection may bias toward greater prospective model-based control than both comparison stimuli. Alternatively, it is possible that valence-context asymmetry increases perceived difficulty resulting in increased model-free contributions as a form of learned helplessness [10].

In the current set of three preregistered studies ($N$ = 600 total), we applied computational modeling to characterize learning for positively-valenced stimuli in disparate contexts (protection and reward; Study 1 and 2) and learning for disparately-valenced stimuli in the same context (protection and punishment; Study 3). We developed five modified versions of a widely-used two-step reinforcement learning task examining protection acquisition (positive valence, aversive context), reward acquisition (positive valence, positive context), and punishment avoidance (negative valence, aversive context) (Fig 1A) [19]. Protection was equated to reward in that both were appetitive stimuli, but the relevance of acquiring each differed by context such that protection reduced negative outcomes (bonus reduction) whereas reward increased positive outcomes (bonus increase). Punishment was negatively valenced such that it was an aversive stimulus to be avoided and increased negative outcomes (bonus reduction). We hypothesized that the context-valence asymmetry for protection would increase model-based contributions compared to reward (Study 1 and 2) in line with prior work showing that even unrelated aversive contexts can interfere with learning for positive outcomes [20]. We hypothesized that model-based contributions for protection would also be higher compared to punishment avoidance given the potential for combined contributions of appetitive and aversive motivations for protection (Study 3). We examined effects of incentives (high versus low stakes) to determine whether differences in model-based control were modulated by incentive [19,21]. We hypothesized that incentive sensitivity would be higher for reward given lower value thresholds for protection and punishment. Lastly, we examined metacognitive and predictive bias on each task to determine how difference in subjective confidence and task performance monitoring related to model-based control and anxiety. We hypothesized increased model-based control would be associated with reduced metacognitive bias (improved correspondence between confidence and performance) across all stimuli.

## Results

In each pre-registered study, a balance between model-free and model-based control was assessed using two variants of a two-step reinforcement learning task. Each study included a protection acquisition variant and either a reward acquisition (Studies 1 and 2) or punishment avoidance comparison (Study 3). During each task, subjects made sequential decisions that navigated them through two "stages" defined by different stimuli. Subjects were told at the outset that they were traveling through a fictious forest. On each trial, subjects were first presented with an indication of whether the trial was a high-stakes or low-stakes trial. High-stakes trials were 5x more valuable than low-stakes trials, as indicated by 1 or 5 flames (protection and punishment variants) or 1 or 5 coins (reward variants). The stakes manipulation was designed to test whether model-based control was modulated by incentive and only included a single low- and high-stakes value (1 and 5, respectively). After the stakes depiction, subjects were shown one of two first-stage states. Each first-stage state included 2 dwellings and subjects chose one dwelling to visit (left or right). In the protection variant dwellings were trees, and in the reward and punishment variants dwellings were houses. First-stage dwellings were randomly presented in 2 equivalent states such that dwellings remained in their pairs throughout but the position of each dwelling (left versus right) was counterbalanced across trials. In total, 4 total dwellings were available for each task variant. In each of the first-stage states, one dwelling led to one creature and the second dwelling led to a different creature (2 total creatures), creating an implicit equivalence across first-stage states. Dwelling-creature pairings remained constant (deterministic transitions). Each second-stage creature was associated with a fluctuating outcome probability. Across all protection task variants, the second-stage outcomes were protection stimuli that reduced losses (shields to protect against the dragon flames). In the Study 1 reward task variant, the second-stage outcomes were reward stimuli that increased gains (sacks to carry the fairy coins out of the forest). In the Study 2 direct reward task variant, the second-stage outcomes were directly delivered as reward stimuli (fairy coins) that increased overall gains. In the Study 3 punishment task variant, the second-stage outcomes were directly delivered as punishment stimuli (dragon flames) that increased overall losses. Second-stage probabilities changed slowly over time, requiring continuous learning in order select the appropriate first-stage state that led to the second-stage creature that provided the most optimized outcome. At the final frame, second-stage outcomes (shields, sacks, coins, flames) were multiplied by the initial stakes to compute an overall point result for that trial, which affected the subject's bonus payment.

To quantify the computational mechanisms underpinning behavior, we fit four computational models to subjects' choice data during the task (see "Methods" for a full description of all models). The parameter of primary interest was the balance between model-free and model-based control between task variants. At an individual level, this balance can be quantified by a hybrid model, which combines the decision values of two algorithms according to a weighting factor ($\omega$). A learning rate ($\alpha$) parameter was also estimated for integrating outcomes to update choice behavior. Additional model-parameters included a single eligibility trace ($\lambda$), stickiness ($\pi$) and inverse-temperature ($\beta$) parameter. Watanabe-Akaike Information Criterion (WAIC) scores were used as a complexity-sensitive index of model fit to determine the best model for each study. WAIC estimates expected out-of-sample-prediction error using a bias-corrected adjustment of within-sample error, similar to Akaike Information Criterion (AIC) and Deviance Information Criterion (DIC)[22]. In contrast to AIC and DIC, WAIC averages over the posterior distribution rather than conditioning on a point estimate, which is why WAIC was selected as the index of model fit.

To interrogate the computational modeling analyses, we used mixed-effects logistic regressions. We present computational results first, followed by mixed-effects regression results consistent with prior work[19] and because the computational results are of primary focus here. Regression was used to examine choice behavior as a function of the outcome on the previous trial and similarity in first-stage state. Choice behavior was measured as the probability of repeating a visit to the same second-stage state ("stay probability"). The interaction between first-stage state and previous outcome indicates a model-free component whereas a main effect of previous outcome indicates a model-based component [23]. For the model-free strategy, outcomes received following one first-stage state should not affect subsequent choices from a different first-stage state because an explicit task structure is not mapped (thus the equivalence between first-stage states is not learned). The model-free learner only shows increased stay probability when the current first-stage state is the same as the first-stage state from the previous trial, and this is reflected as an interaction between previous outcome and first-stage state. The model-based learner, in contrast, uses the task structure to plan towards the second-stage outcomes, allowing it to generalize knowledge learned from both first-stage states. Thus, outcomes at the second stage equally affect first-stage preferences, regardless of whether the current trial starts with the same first-stage state as the previous trial.

## Study 1: Comparing protection acquisition to reward acquisition

In Study 1, two-hundred subjects ($M_{age}$ = 27.99(6.87), range$_{age}$ = 18–40 years, 98 females, 102 males) completed the protection acquisition and reward acquisition task variants (Fig 1B and 1C). The aim in both variants was to earn the maximum possible second-stage outcome thereby optimizing the final result. Second-stage outcomes for the protection acquisition variant were shields that served as protective stimuli to reduce punishment in the form of dragon flames. Second-stage outcomes for the reward acquisition variant were sacks that served as reward stimuli to increase reward in the form of fairy coins (Fig 1C). The number of second-stage outcomes earned ultimately affected the final result, which was points that contributed to subjects' bonus payments.

## Task engagement

Subjects were engaged and performed well, as shown by higher than median available outcomes earned (S1A Fig). Average number of points earned per trial (reward rate) was calculated for each subject and mean corrected by the average outcome available to them via individually generated point distributions. Corrected reward rate was higher on the protection task variant and did not differ as a function of stakes within each task variant (S1B Fig). Subjects made first-stage decisions in less than 1 second, and first-stage reaction time (RT) was significantly faster for the protection variant (S1C Fig).

## Computational models

The best fitting model was Model 3, which included separate model-based weighting ($\omega$) and learning rate ($\alpha$) parameters for each task variant, as well as eligibility trace ($\lambda$), stickiness ($\pi$) and inverse-temperature ($\beta$) parameters (S1 Table). Subjects had higher average $\omega$ on the protection variant ($M$ = .72, $SD$ = .14) compared to the reward variant ($M$ = .14, $SD$ = .22), $t(199)$ = 17.43, $p < .001$, 95% CI [.27, .34] (Fig 2A). $\alpha$ did not significantly differ by task variant (protection $M$ = .47, $SD$ = .33; reward $M$ = .43, $SD$ = .32; $t(199)$ = 1.59, $p$ = .11, 95% CI [-.01, .08]) (Fig 2B). $\omega$ and $\alpha$ were positively correlated, protection $r(199)$ = .26, $p < .001$; reward $r(199)$ = .33, $p < .001$.

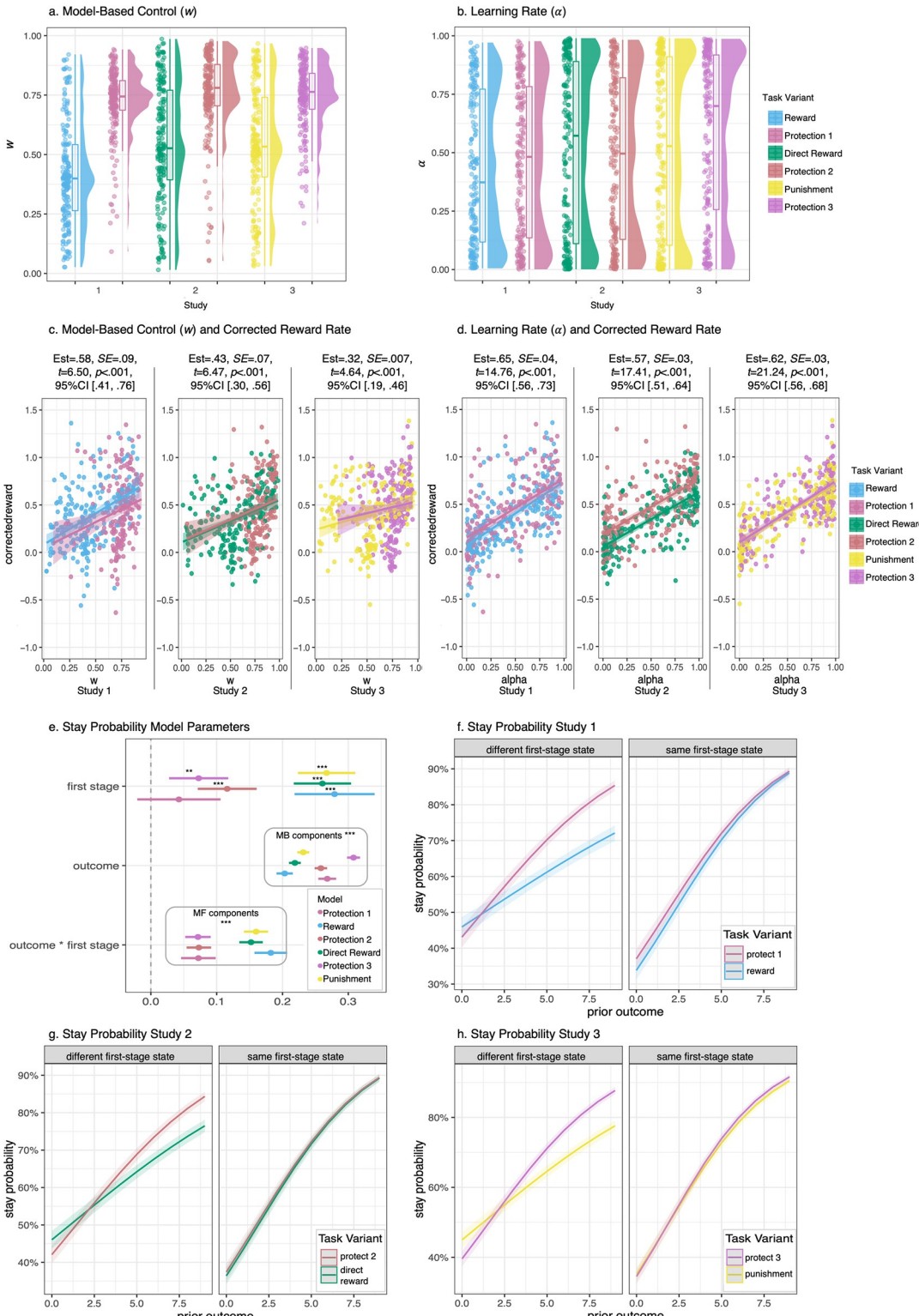

**Fig 2. Model-Based Control and Learning Rate Results. a-b.** Raincloud plots depicting model-based control weighting ($\omega$) and learning rate ($\alpha$) by study and task variant. $\omega$ was significantly higher for the protection variants compared to all other task variants. $\alpha$ did not significantly differ across task variants. Far right legend indicates task variants across all studies: Study 1 = Reward and Protection 1, Study 2 = Direct Reward and Protection 2, Study 3 = Punishment and Protection 3. **c-d.** Scatterplots and linear regression lines depicting positive associations between both $\omega$ and $\alpha$ with corrected reward rate by

study and task variant. Higher $\omega$ and $\alpha$ were significantly associated with higher corrected reward rate for all task variants. **e.** Mixed-effects model parameters testing contributions of the first-stage state, prior trial outcome, and interaction between first-stage state and prior trial outcome on stay probabilities. Effects from both model-free and model-based contributions were observed. MF = model-free control; MB = model-based control. **f-h.** Mixed-effects models testing model-based (different first-stage state) and model-free (same first-stage state) contributions to stay probabilities (likelihood of repeating the same second-stage state). Increased model-based contributions were revealed on the protection task variants compared with all other task variants.

To confirm a higher degree of model-based decision making led to better performance, we examined associations between corrected reward rate and $\omega$, controlling for task order and variant. Higher $\omega$-value led to better performance, (Fig 2C). Higher $\alpha$ was also associated with better performance (Fig 2D). No moderation by task variant was observed. Accounting for correlation between $\omega$ and $\alpha$ did not change effect on corrected reward rate, $\omega$ Est = .30, $SE$ = .08, $t$ = 3.86, $p < .001$, 95% CI [.15, .46], $\alpha$ Est = .59, $SE$ = .05, $t$ = 13.22, $p < .001$, 95% CI [.51, .68].

## Mixed-effects models

A significant main effect of previous outcome on staying behavior was observed, indicating contributions from model-based control, Est = .23, $SE$ = .005, $z$ = 50.96, $p < .001$, 95% CI [.22, .24] (Fig 2E). An interaction between first-stage state and previous outcome was also significant, indicating contributions from a model-free component, Est = .13, $SE$ = .009, $z$ = 14.11, $p < .001$, 95% CI [.11, .15]. Lastly, a three-way interaction with task variant was observed confirming reinforcement learning results that subjects were more model-based on the protection task variant. In other words, prior outcome had a stronger effect on stay behavior for the protection variant compared to the reward variant when the first-stage states differed from the previous trial, but not when the first-stage state was the same, Est = -.11, $SE$ = .02, $Z$ = -6.23, $p < .001$, 95% CI [-.15, -.08] (Fig 2F).

Reinforcement learning models did not reveal a meaningful stakes effect (Model 4 fit was not significantly better than Model 3). Stakes also did not moderate stay behavior with either the model-based or model-free mixed-effects components (S2A Fig). These results were consistent for each task variant individually. Because this null effect of stakes differed from prior work using a similar task and our task included fewer trials compared with prior work [21], we explored whether the effect of stakes varied as a function of task duration. Task duration interacted with stakes and previous outcome, such that there was no effect of stakes at the start of the task but high-stakes trials had an increase in likelihood of stay behavior at the end of the task, Est = -.001, $SE$ = .0003, $Z$ = -2.10, $p$ = .036, 95% CI [-.001, -.00004] (S2B Fig).

## Study 2: Comparing protection acquisition to direct reward

In Study 2, two-hundred subjects ($M_{\text{age}}$ = 23.26(4.02), range$_{\text{age}}$ = 18–36 years, 152 females, 48 males) completed the protection acquisition and direct reward acquisition task variants (Fig 1B and 1D). Study 2 was conducted with the same protection acquisition task as Study 1, but substituted the reward acquisition task with a modified version to test whether indirect reward delivery in Study 1 (i.e., learning to acquire sacks rather than coins themselves) artificially reduced model-based control contributions to reward acquisition (Fig 1D). Study 2 also increased the number of trials for both task variants. Study 1 did not reveal an effect of stakes, as found in prior work examining reward learning [21], but Study 1 had a fewer number of trials than prior work. We thus tested whether the lack of stakes effect in Study 1 was due to the number of trials by increasing the number of non-practice trials in Study 2 from 100 to 200 for each variant, consistent with prior work [21]. Again, the aim in both task variants was to earn the maximum possible outcome (shields, coins) thereby optimizing the final result (points contributing to bonus).

### Task engagement

As in Study 1, subjects were engaged and performed well, with higher than median outcomes earned, RTs of <1 second, and higher corrected reward rate for protection compared to direct reward (S1A–S1C Fig). No stakes effect was observed for corrected reward rate.

### Computational models

Computational Model 4 was the best fitting model for Study 2, which included separate $\omega$ and $\alpha$ parameters for task variant and stakes (S3 Fig). However, we report Model 3 results which also fit well to improve comparability across studies. $\omega$ was higher for the protection variant ($M = .76$, $SD = .18$) compared to the direct reward variant ($M = .55$, $SD = .25$), $t(199) = 11.00$, $p < .001$, 95% CI [.17, .25], consistent with Study 1 (Fig 2A). As in Study 1, $\alpha$ did not significantly differ by task variant (protection $M = .48$, $SD = .35$; direct reward $M = .51$, $SD = .37$; $t(199) = -1.23$, $p = .22$, 95% CI [-.09, .02]) (Fig 2B). $\omega$ and $\alpha$ were positively correlated, protection $r(199) = .25$, $p < .001$; direct reward $r(199) = .38$, $p < .001$. See S1 Table for $\lambda$, $\pi$, and $\beta$ parameters. Consistent with Study 1, higher $\omega$ and $\alpha$ parameters were associated with better performance as indexed by increased corrected reward rate, with no moderation by task variant (Fig 2C and 2D). Accounting for correlation between $\omega$ and $\alpha$ did not change effect on corrected reward rate, $\omega$ Est = .16, $SE = .06$, $t = 2.81$, $p = .005$, 95% CI [.05, .26], $\alpha$ Est = .54, $SE = .03$, $t = 15.74$, $p < .001$, 95% CI [.47, .61].

### Mixed-effects models

Mixed-effects models identified model-based, Est = .24, $SE = .003$, $z = 72.62$, $p < .001$, 95% CI [.23, .24], and model-free contributions, Est = .11, $SE = .006$, $z = 17.11$, $p < .001$, 95% CI [.09, .12] (Fig 2E), and replicated the moderation by task variant observed in Study 1, Est = -.08, $SE = .01$, $z = -5.92$, $p < .001$, 95% CI [-.10, -.05] (Fig 2G). Stakes interacted with the model-based component, which was driven by the direct reward variant (S2C Fig). No significant stakes interaction was present for the model-free component.

### Study 3: Comparing protection acquisition to punishment avoidance

In Study 3, two-hundred subjects ($M_{age} = 22.25(3.85)$, $range_{age} = 18$–$39$ years, 159 females, 41 males) completed the protection acquisition and punishment avoidance task variants (Fig 1B and 1E). The longer protection acquisition variant from Study 2 was used. The punishment avoidance variant was a traditional aversive avoidance variant where the aim was to avoid punishment (dragon flames) that was delivered directly at stage-two (Fig 1E).

### Task engagement

Outcomes earned were higher than median available outcomes, RTs were <1 second, and subjects earned more for protection compared to punishment avoidance (S1A–S1C Fig). No stakes effect was observed for corrected reward rate. RT for first-stage decisions differed by stakes, such that RTs were slower for high stakes trials (S4 Fig).

### Computational models

As in Study 1, the best fitting model was Model 3, which included separate $\omega$ and $\alpha$ parameters for task variant, but not for stakes type. Subjects had higher $\omega$ parameters on the protection variant ($M = .74$, $SD = .14$) compared to the punishment avoidance variant ($M = .54$, $SD = .25$), $t(199) = 10.94$, $p < .001$, 95% CI [.17, .24] (Fig 2A). $\alpha$ was significantly higher for protection compared to punishment avoidance, revealing the only learning-rate difference by task

variant across the studies (protection $M = .58$, $SD = .35$; direct reward $M = .51$, $SD = .37$; $t(199)$ = 2.74, $p = .007$, 95% CI [.02, .12]) (Fig 2B). $\omega$ and $\alpha$ were positively correlated for punishment avoidance, $r(199) = .15$, $p = .04$, but not protection $r(199) = .10$, $p = .16$. See S1 Table for $\lambda$, $\pi$, and $\beta$ parameters. Consistent with Studies 1 and 2, higher $\omega$ and $\alpha$ led to better performance, as indexed by corrected reward rate, with no moderation by task variant (Fig 2C and 2D). Accounting for correlation between $\omega$ and $\alpha$ revealed no significant association with corrected reward rate, $\omega$ Est = .17, $SE = .05$, $t = 3.30$, $p = .001$, 95% CI [.06, .27], $\alpha$ Est = .61, $SE = .03$, $t = 20.76$, $p < .001$, 95% CI [.55, .67].

## Mixed-effects models

Mixed-effects models identified model-based, Est = .26, $SE = .003$, $z = 79.23$, $p < .001$, 95% CI [.26, .27], and model-free contributions, Est = .12, $SE = .007$, $z = 17.44$, $p < .001$, 95% CI [.10, .13] (Fig 2E), and replicated the moderation by task variant observed in Studies 1 and 2, Est = -.09, $SE = .01$, $z = -6.42$, $p < .001$, 95% CI [-.11, -.06] (Fig 2H). No stakes effect was observed with respect to either model-based or model-free component (S2D Fig).

## Computational model parameter comparisons between studies

Model-based control ($\omega$) for the protection variant did not significantly differ across studies, $F(2, 597) = 2.56$, $p = .08$. For non-protection task variants (i.e., reward and punishment avoidance), $\omega$ differed across studies, $F(2, 597) = 18.93$, $p < .001$. Post-hoc Tukey HSD comparisons revealed a significant difference between reward in Study 1 and direct reward in Study 2 as well as punishment avoidance in Study 3, such that both Study 2 and 3 $\omega$ were .13 higher than Study 1, $p < .001$. Direct reward in Study 2 and punishment avoidance in Study 3 did not differ, $w_{diff} = -.004$, $p = .99$.

Learning rate ($\alpha$) significantly differed across studies for the protection task variants, $F(2, 597) = 7.07$, $p = .001$, as well as non-protection task variants, $F(2, 597) = 3.76$, $p = .02$. Post-hoc Tukey HSD comparisons revealed a significant difference between protection in Study 3 and protection in Study 1 as well as protection in Study 2, such that Study 3 $\alpha$ was .12 higher than Study 1, $p = .002$, and .007 higher than Study 2, $p = .007$. Study 1 reward $\alpha$ was .09 lower than Study 3 punishment, $p = .04$.

## Metacognitive and predictive bias

Model-based actions can be implicit, where there is a nonconscious anticipation of an outcome, or explicit, where a conscious prospection can motivate behavior [24]. We assessed measures of metacognitive (certainty) and predictive bias (outcome estimates) to examine whether subjective evaluation tracked outcomes and whether bias differed as a function of learning strategy and context.

Average certainty and outcome estimate ratings on a scale of 0–9 were above the midpoint for all task variants (Fig 3A and 3B). Mixed effects regression results indicated that certainty and outcome estimates were higher on trials for which subjects earned higher outcomes, reflecting metacognitive and predictive bias, respectively (Fig 3C).

Random slope coefficients were extracted from the model of outcome predicting certainty and outcome estimates. Coefficients represented metacognitive and predictive bias for each subject. Metacognitive bias only differed by task variant for Study 3, with reduced bias for protection than punishment avoidance (Fig 3D). Predictive bias was lower for protection than reward and punishment in Studies 1 and 3, but higher for direct reward compared to protection in Study 2 (Fig 3E). Bias coefficients were regressed against $\omega$ and $\alpha$ for each task variant. Reduced metacognitive bias was associated with faster learning rate for all tasks and more

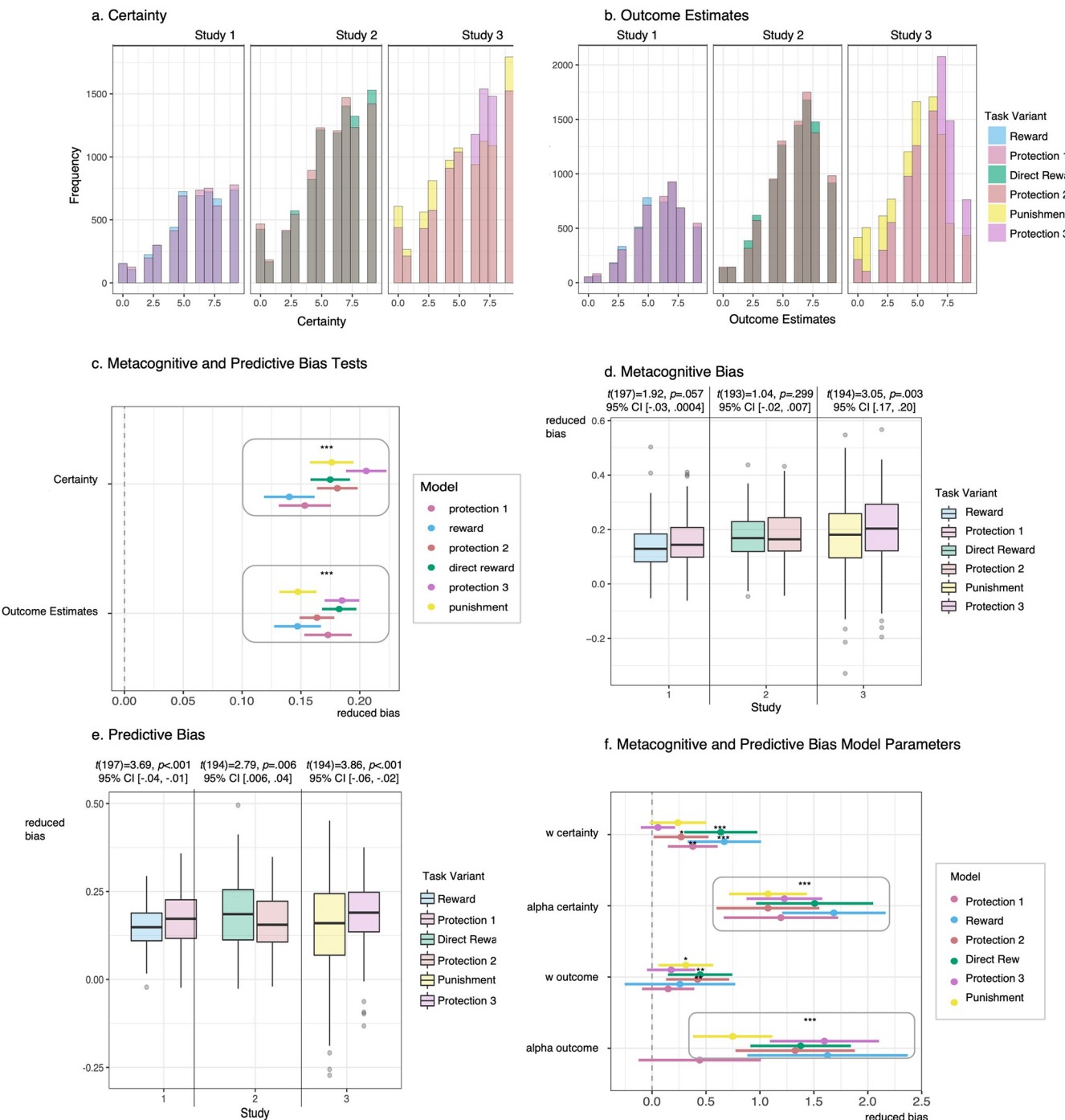

**Fig 3. Metacognitive and predictive bias results. a.** Histograms depicting Certainty ratings by study and task variant. Certainty was rated with respect to how sure subjects felt they were that they selected the first-stage state that would lead to the most optimal outcomes. Certainty ratings were made on a scale of 0–9 from not at all certain to very certain. **b.** Histograms depicting Outcome Estimates by study and task variant. Outcome Estimates were provided with respect to how many outcome units subjects thought they would receive at the second-stage. Outcome Estimates were made on a scale of 0–9 outcome units (i.e., subjects who rated a 2 thought they would receive 2 shields/sacks/coins/flames, respectively). **c.** Mixed-effects model parameters testing metacognitive and predictive bias by modeling actual outcome received as a function of Certainty and Outcome Estimates, respectively. **d.** Metacognitive bias boxplots by study and task variant. Metacognitive bias was calculated by extracting random slope coefficients from the model of outcome predicting Certainty. Significant differences were only identified in Study 3 with reduced bias for the protection acquisition variant compared to the punishment avoidance variant. **e.** Predictive bias boxplots by study and task variant. Predictive bias was calculated by extracting random slope coefficients from the model of outcome predicting Outcome Estimates. Significantly reduced bias was revealed for the protection acquisition variants compared to the reward acquisition and punishment avoidance variants, but not compared to the direct reward variant. **f.** Model parameters for metacognitive and predictive bias coefficients regressed against model-based control weighting

($\omega$) and learning rate ($\alpha$) parameters for each task variant. Far right legends indicate task variants across all studies: Study 1 = Reward and Protection 1, Study 2 = Direct Reward and Protection 2, Study 3 = Punishment and Protection 3.

model-based control for protection and reward in Studies 1 and 2 (Fig 3F). Predictive bias was inconsistently related to model-based control, but was related to slower learning rate across all task variants with the exception of protection in Study 1 (Fig 3F).

## Anxiety

All subjects provided self-report assessments of anxiety using the State-Trait Anxiety Inventory (STAI) Trait Anxiety subscale [25]. Average scores were $M$ = 29.59, $SD$ = 11.37, range = 1–58 across all studies. Differences in deployment of model-based control ($\omega$-difference score calculated as non-protection variant subtracted from the protection acquisition variant for each study) were associated with anxiety such that individuals with higher scores on the STAI demonstrated greater model-based weighting for reward acquisition compared with protection acquisition, but greater model-based weighting for protection acquisition compared with punishment avoidance: study by $\omega$-difference interaction Estimate = 5.40, $SE$ = 2.21, $t$ = 2.45, $p$ = .015, 95% CI [1.07, 9.74], $R^2$ = .02 (Fig 4). Anxiety was not significantly associated with learning rate differences: Estimate = 1.38, $SE$ = 1.66, $t$ = .83, $p$ = .406, 95% CI [-1.88, 4.63], $R^2$ = .01.

For certainty, anxiety interacted with task type such that individuals with higher anxiety reported reduced certainty for protection acquisition compared to reward and direct reward, but increased certainty for protection acquisition compared to punishment avoidance (S5A Fig). For outcome estimates, subjects with higher anxiety also demonstrated lower outcome estimates for protection acquisition compared with direct reward, but higher outcome estimates for protection acquisition compared to punishment avoidance (S5B Fig). Anxiety was not significantly associated with metacognitive or predictive bias.

## Discussion

This is the first study we are aware of that determines how humans apply reinforcement learning strategies to adaptively acquire protection. Reinforcement learning models describe how predictions about the environment facilitate adaptive decision making. In aversive contexts, predictions center on minimizing harm whereas appetitive contexts motivate reward maximization. Traditional safety conceptualizations center on threat and, as such, typically elicit avoidance as opposed to approach behavior [26]. However, the current study required approach behavior to maximize positively-valenced protection. Our results demonstrate that individuals engaged model-based control systems to a greater extent when acquiring protection compared to acquiring reward and avoiding punishment. By reconceptualizing safety in terms of appetitive protection, this study progresses understanding of context-valence interactions underlying differential recruitment of decision control systems.

In contrast to prior studies that consider safety in terms of punishment avoidance, the current studies aligned protection with reward by making both positively valenced (i.e., more is better). We compared protection with reward to determine whether there was something conceptually different about the way individuals learn for these types of stimuli, or whether protection is simply reward by a different name. Our results suggest the former. We interpret these findings to suggest that, despite similar valence, protection acquisition is conceptually different from other types of reward. The ultimate goal of protection acquisition is to minimize harm, whereas reward acquisition does not explicitly consider harm. During value-based choice, individuals first assigning values to all of the stimuli that can be obtained, and then compare

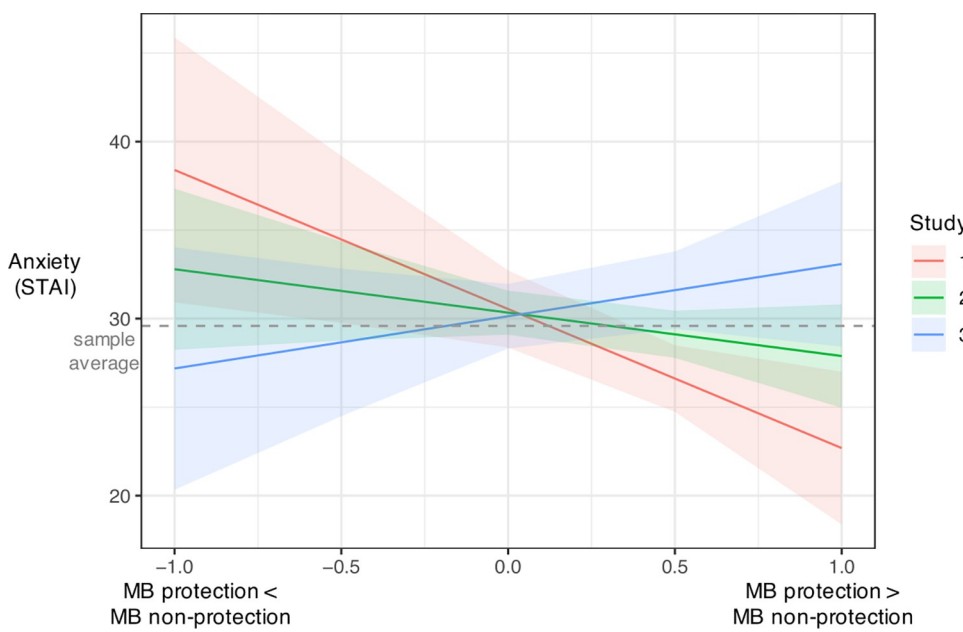

**Fig 4. Anxiety and model-based weighting ($\omega$) estimated separately for each Study.** Model-based prioritization was observed for protection compared with punishment avoidance (Study 3) for individuals with higher anxiety scores and for protection compared with reward acquisition for individuals with lower anxiety scores (Study 1 and 2). Dashed grey line represents the sample average scores on the State-Trait Anxiety Inventory (STAI) Trait Anxiety subscale.

the computed values to select one. In real-world contexts, multiple value-based choices that span appetitive and aversive outcomes may occur simultaneously. For example, perhaps you build a fence with a friend because dangerous mountain lions invade your yard. But while building the fence, you also have positive experiences like sharing time with a friend and drinking a refreshing lemonade. Importantly, the friend and lemonade may be positive but are not protective stimuli. Instead, they are other rewarding social stimuli that co-occur with protection acquisition. Thus, the value-based choice of building a fence versus digging a trench is dissociable from the value-based choice of which friend to invite to help you or the choice of whether to have iced tea or lemonade. The current experimental tasks were designed to disentangle appetitive and aversive motivation with respect to the type of outcome faced (reward or loss), as is standard with examinations of appetitive/aversive domains [27,28]. Using both computational modelling and model-agnostic analyses, our findings revealed that protection amplifies contributions from the model-based system when compared with traditional appetitive reward and aversive punishment.

The current tasks were designed such that greater model-based control was yoked to more beneficial outcomes through deterministic state transitions [23]. This feature clarifies prior work that suggested greater reliance on model-based control may be suboptimal in aversive contexts [11]. When the goal is to maximize protection, and deployment of model-based control can better achieve that goal, individuals show increased reliance on model-based control. Thus, aversive context can motivate flexible and adaptive behavior for positively-valenced outcomes. Differences in model-based control were not attributable to task complexity as evinced by faster reaction time for protection than other task variants and no significant differences in learning rate between tasks. Thus, it is not that protection as a construct is more abstract thereby requiring more effortful control to optimize behavior. Instead, these results suggest value-based features of protection drive engagement of more flexible, goal-directed learning systems.

Metacognitive and predictive biases were reduced for acquiring protection than avoiding punishment. In line with the reinforcement learning results, it is likely that punishment avoidance engages reflexive decision circuits, whereas protection acquisition more closely approximates distal threat affording engagement of cognitive circuits [8]. These findings clarify prior work identifying an increase in metacognitive bias in aversive contexts [29], and provide support for our assertion that approach and avoidance motivations underlie differences previously attributed to context. Because our design improved accuracy-demand trade-offs through deterministic state transitions, our results are also consistent with recent work showing higher decision confidence is associated with model-free learning when model-based and model-free systems have chance level performances [30]. In this study, reduced metacognitive bias was associated with more model-based control and faster learning rates. Thus, model-based actions can be interpreted to reflect explicit forecasts. Predictive accuracy demonstrated the same general pattern with less consistency, perhaps because of the inherent difficulty in estimating precise outcomes with random walks. Together, metacognitive bias findings suggest that a boost in model-based control by reframing safety as an approach toward protection mitigates metacognitive differences previously linked to context.

Individual differences in trait anxiety were associated with degree of model-based control deployed to acquire protection, offering a potential mechanistic explanation for differences in safety decisions previously documented in anxious individuals [31]. For individuals with higher anxiety, model-based control for protection was decreased compared with reward. In a separate sample, model-based control was elevated for protection compared with punishment. This increase in model-based control depending on valence-context interactions also supports our assertion that protection acquisition is distinct from purely aversive punishment and appetitive reward. Trait anxiety was also associated with a general reduction in certainty and outcome estimates across the valence spectrum, but not with increased metacognitive or predictive bias. Together, these finding suggests that individuals with higher anxiety performed worse on tasks involving negative context, but that they were able to estimate their performance with comparable accuracy to those with lower anxiety scores. The increase in model-based control for protection compared to punishment has implications for real-world behaviors observed in anxiety. Model-free responses to protection acquisition can lead to repeating overly-cautious avoidance behaviors, often referred to as problematic "safety-seeking". Model-based control, however, can facilitate flexible updating in response to changing threat contingencies, which supports adaptive safety acquisition. The increase in strategic control for protection compared with punishment raises the possibility that leveraging approach motivation may be beneficial for protection-based learning in anxious individuals. Applying computational decision frameworks to safety extends understanding of how humans rationally acquire protection in the face of threat and how decision control strategies differ compared with other appetitive and aversive stimuli.

This study has implications beyond informing theoretical frameworks to potentially expanding clinical approaches to treating anxiety. Up to 50% of individuals with anxiety do not fully respond to current treatments (e.g., cognitive behavioral therapy) [32]. This may be, in part, due to clinical science conceptualizations of safety seeking as dysfunctional avoidance [33,34] contributing to the onset and maintenance of anxiety [35]. However, recent work proposes focusing on learning about safety, as opposed to threat, may be a promising alternative avenue by which to improve anxiety treatment [36,37]. The current studies supports this call to disaggregate threat extinction and safety acquisition. Our findings show adaptive safety acquisition does not function the same as threat avoidance, even for those with higher trait anxiety. Similar to conditioned inhibition approaches, our findings indicate safety via protection can be trained in the presence of threat thereby reducing competing associations formed

during Pavlovian extinction learning [36]. The current paradigm also has the potential to differentiate between maladaptive coping strategies such as avoidance and adaptive coping strategies[38] such as flexible safety maximization by examining decisions to acquire protection when doing so is rational and in the presence of threat.

Study findings should be considered in the context of design limitations. The sample was recruited and tested online without a primary aversive stimulus (i.e., shock). Monetary outcomes have been previously validated for studying aversive and appetitive learning [16,19,24], but other aversive contexts will need to be tested to enhance generalizability. Although we based our paradigm development on widely-used and validated reinforcement-learning tasks, we only replicated the stakes effect observed in prior work in Study 2 [21]. The model accounting for both task variant and stakes fit best for Study 2, but the WAIC score for the more complex model was only .18% different from the simpler model, thus we used Model 3 to compare across studies. Despite prior work identifying higher exploit behavior under high-stakes [19], we did not test an inverse-temperature difference by stakes considering the lack of stakes effect on model-based weighting and the potential for non-identifiability given inverse-temperature interacts multiplicatively with the weighting parameter [39]. The lack of a robust effect of incentives (i.e., stakes) raises the possibility that other task-based factors not yet classified in the reinforcement-learning literature are at play [23]. Additional tasks exploring approach-based safety are needed to further validate and replicate the constructs examined here. We did not examine working memory effects, which have been recently argued to be relevant for performance on two-step tasks [40]. We did not collect race and ethnicity data for our sample, which precludes conclusions as to whether our sample adequately represents the broader population. We also did not assess intolerance of uncertainty, which is considered a lower order factor related to anxiety, and is often correlated with trait anxiety, but has independent predictive value [41]. Intolerance of uncertainty has been shown to specifically relate to physiological regulation in response to threat and safety cues during conditioning [42,43]. With regard to decision control systems, prior findings suggests model-free control may be more optimal under high uncertainty, particularly for punishment avoidance [11]. However, more work is needed to understand how dispositional intolerance of uncertainty interacts with situational uncertainty to influence decision control systems and learning.

Every day, humans seek to acquire protection through prospective decisions, which engage model-based decision control systems. The current studies illuminate computational decision control components that differentiate protection acquisition from reward acquisition and punishment avoidance. We focus on how humans make adaptive decisions to seek out protective stimuli as a rational choice behavior when threat is present. This focus on beneficial safety seeking differs from examinations of aberrant safety seeking, which currently dominate the literature given important ties to psychopathology. Here we provide evidence that the valence and context asymmetry of protection increased goal-directed control compared with other stimuli that have consistent valence and context matching (i.e., reward and punishment). By identifying the engagement of flexible decision control systems in protection acquisition, this work lays the foundation for better understanding of how humans adaptively acquire safety and how safety learning goes awry in psychopathology.

## Materials and methods

### Ethics statement

All methodology was approved by the California Institute of Technology Internal Review Board, and all subjects provided written consent to participate through an online consent form at the beginning of the experiment. Subjects were compensated for their time at a rate of

US$9.00 per hour and were entered into a performance-contingent bonus lottery for US $100.00. The lottery served to increase task engagement.

## Sample

Six-hundred human subjects completed two reinforcement learning tasks online across three studies. Each study involved an independent sample of 200 subjects. Study 1 compared a protection and reward variant (Fig 1B and 1C), Study 2 compared a longer version of the protection variant and a direct reward variant (Fig 1B and 1D), and Study 3 compared the Study 2 protection variant and a punishment avoidance variant (Fig 1B and 1E). Subjects were recruited through Prolific, an online recruitment and data collection platform that produces high-quality data [44]. As described in our preregistration, we used a stopping rule of 200 subjects with useable data. In each Study, subjects completed two task variants (Study 1, 90-minutes; Studies 2 and 3, 120-minutes).

## Inclusion and exclusion criteria

Subjects were included based on being aged 18–40, fluent in English, and normal or corrected vision. Subjects were excluded from all analyses and replaced through subsequent recruitment if they failed to respond to more than 20% of trials within the allotted time or if they incorrectly responded to more than 50% of comprehension checks. In total, 15 subjects (2.5% of the total sample) failed these criteria (Study 1, 3; Study 2, 8; Study 3) and were replaced through subsequent recruitment.

## Materials and procedure

In Study 1, subjects played two similar games in which they were traveling through a fictitious forest with a goal to either maximize protection (protection variant) or reward (reward variant). Each variant consisted of 120 trials, with the first 20 trials designated as practice and not included in analyses. In Study 2, subjects played a longer version of the same protection variant and a modified version of the reward variant with reward directly delivered at Stage 2 (direct reward variant). In Study 3, subjects played the longer protection variant and a punishment variant with the goal to minimize punishment (punishment avoidance). In Study 2 and 3, trial numbers were increased to 225 trials, with the first 25 designated as practice in line with prior work examining the effect of stakes [21]. Presentation of the two task variants were counterbalanced across subjects for each study.

Prior to completing the tasks, subjects were instructed extensively about the transition structure, outcome distribution, and how the stakes manipulation worked. Subjects completed 10 comprehension questions (no time limit) with feedback after the task instructions. Subjects were excluded from analyses and replaced through subsequent recruitment if they completed less than 50% of comprehension questions accurately. Instructions and comprehension were included to ensure subjects fully understood task elements.

## Stakes

Each trial started randomly with an indicator of high (x5) or low (x1) stakes for 1500ms. Specifically, in all protection and the punishment avoidance variants subjects were shown dragons who were either small and delivered one flame or were large and delivered five flames. In the reward and direct reward variants subjects were shown fairies that were either small and delivered one gold coin or were large and delivered five gold coins. Low-stakes results ranged from 0–9 units of reward/punishment whereas high-stakes results ranged from 0–45 units. This

allowed for an outcome/reward distribution similar to that used previously with varying stakes magnitudes [19]. The stakes amplifier was applied to the punishment/reward available on that trial as well as the final result received.

### First-stage choices

After the stakes depiction, one of two possible first-stage states was randomly shown. In all protection variants, first-stage states were depicted as trees where gnomes dwelled. In all other variants, first-stage states were depicted as houses where elves dwelled. Houses were used to denote reward, direct reward, and punishment in order to minimize between-study differences in comparisons as a function of stimulus kind. Subjects had to choose between the left- and right-hand first-stage dwellings using the "F" and "J" keys within a response deadline of 1500ms. If subjects did not respond within the time allotted, they were told they did not select in time and were instructed to press "space" to start the next trial.

### Second-stage outcomes

First-stage choices determined which second-stage state was encountered. Deterministic transitions specified that the same first-stage dwelling always led to the same second-stage state, which was depicted as a creature. Choices between first-stage states were equivalent between such that a dwelling in each pair always led to one of the two second-stage creatures and the other always led to the remaining creature. This equivalence distinguished model-based and model-free strategies because only the model-based system can transfer learned experiences from one first-stage state to the other [23]. This is an important aspect of the task given growing evidence that both model-free and model-based strategies can result in optimal decisions depending on task constraints [45]. In the Study 1 reward variant, second-stage creatures were elves who made sacks to carry the fairy's gold coins. In the Study 2 direct reward and Study 3 punishment avoidance variants, coins and flames were delivered at the second-stage. In all protection variants, second-stage creatures were gnomes who made shields to protect against the dragon's flames. Payoffs were initialized as low (0–4 points) for one creature and high (5–9 points) for the other. Payoffs changed slowly over the course of the task according to independent Gaussian random walks ($\sigma = 2$) with reflecting bounds at 0 and 9 to encourage learning throughout. A new set of randomly drifting outcome distributions was generated for each subject.

### Final result

After making their first-stage choice, subjects were shown which creature they visited for 1500ms and then were shown how much outcome they received as well as the final result based on how outcomes were applied to the initial stakes for 2500ms. Outcomes were multiplied by stakes and presented as the final result applied to the maximum reward/penalty available on each trial (see Fig 1 for an example). Subjects lost points if attacked in the protection and punishment avoidance variants and gained points in the reward and direct reward variants. Points contributed to actual bonus money distributed. Similar incentives have successfully been used in prior work [12].

### Reinforcement learning models

Reinforcement learning models were fit using a hierarchical Bayesian approach, assuming subject-level parameters are drawn from group-level distributions, implemented in Stan [46], which allowed us to pool data from all subjects to improve individual parameter estimates.

Building on prior work [19,23], we fit 4 computational reinforcement learning models with weighting parameters ($\omega$) that determined the relative contribution of model-based and model-free control and learning-rate parameters ($\alpha$) that governed the degree to which action values were updated after a positive outcome. Models also included an eligibility trace parameter ($\lambda$) that controlled the degree to which outcome information at the second stage transferred to the start stage, a "stickiness" parameter ($\pi$) that captured perseveration on the response, and an inverse-temperature parameter ($\beta$) which controlled the exploitation exploration trade-off between two choice options given their difference in value. Model fitting was conducted as follows: (Model 1) First we fit a "null model" that did not include an effect of stakes or task variant and accounts for subjects' choices by integrating first-stage value assignment for both model-based and model-free systems. Four distinct first-stage states were assumed. (Model 2) Next, we fit a model that included the same first-stage model-based and model-free learning as Model 1 with an additional separate $\omega$ and $\alpha$ parameter for the effect of high- and low-stakes trials. (Model 3) Then, we fit a model that included the same first-stage model-based and model-free learning as Model 1 with an additional separate $\omega$ and $\alpha$ parameter for each task variant. (Model 4) Finally, we fit a model that included the same first-stage learning and task variant effect as Model 3 with an additional separate $\omega$ and $\alpha$ parameter for the effect of high- and low-stakes trials.

Parameters were specified using non-centered parametrizations, whereby each subject-level parameter ($\theta_{subject}$) is formed by a group-level mean ($\mu_{group}$) and standard deviation ($\sigma_{group}$) plus a subject-level offset parameter ($\epsilon_{subject}$):

$$\theta_{subject} = \mu_{group} + \sigma_{group} \cdot \epsilon_{subject}$$

We used weakly informative prior distributions (normal distributions with mean = 0 and standard deviation = 1) on each of these parameters and assigned a lower bound of zero for the standard deviations.

Subject-level parameters $\theta_{subject}$ were subject to logistic sigmoid (inverse logit) transformations to map them into the range [0, 1]. For the inverse temperature parameter, this was multiplied by 20 to give the range [0, 20].

Posterior distributions for model parameters were estimated using Markov chain Monte Carlo (MCMC) sampling implemented in Stan, with 4 chains of 4000 samples each. For further analyses, we used the mean of each parameter's posterior distribution. Model comparison was performed using Watanabe-Akaike Information Criterion (WAIC) scores [47], which provides a goodness of fit measure for Bayesian models penalized according to the number of free parameters in the model. Lower WAIC scores indicate better out-of-sample predictive accuracy of the candidate model. WAIC scores for all models and all studies are reported in S2 Table.

## Mixed-effects models

All statistical tests, with the exception of the reinforcement learning models, were conducted in R (version 4.0.3) using the lme4 package (version 1.1.26) [48]. Mixed effects models were tested using the lmer function (lmerTest assessed t-tests using Satterthwaite's method). Linear models were tested using the lm function. General effects sizes are reported as 95% confidence intervals. Model effect sizes reported as $R^2$ are conditional effects of variance explained by the entire model [49].

## Metacognitive and predictive bias

Decision certainty and outcome estimates were collected throughout the tasks on 25% of trials each. Subjects were asked to report on a scale of 0–9 how certain they were that they selected

the first-stage state that would lead to the best outcome and to estimate the number of outcome units they thought they would receive at the second-stage on a scale of 0–9. Certainty and outcome estimates were not elicited on the same trial.

## Anxiety

As preregistered, anxiety was measured using the State-Trait Anxiety Inventory (STAI) Trait Anxiety subscale [25]. STAI is a 20-item measure on a 4-point scale ranging from "almost never" to "almost always". Trait anxiety evaluates relatively stable aspects of anxiety proneness including general states of calmness, confidence, and security. Internal consistency for anxiety in this sample was Cronbach's $\alpha$ = .94. Individual differences were analyzed with respect to difference in model-based control and learning rate across task variants operationalized as each model parameter for the protection variant minus the corresponding parameter for the non-protection variant within a given study. Positive values reflect increased model-based control and learning rates for the protection variant.

## Preregistration

The main hypotheses and methods were preregistered on the Open Science Framework (OSF), https://osf.io/4j3qz/registrations.

## Supporting information

**S1 Fig. Task performance. a.** Histograms depicting the number of outcome units earned by study and task variant. Black lines indicate median available outcome for each study. **b.** Corrected reward rate boxplots by study and task variant. Corrected reward rate was significantly higher for the protection task variants compared to all other task variants. Corrected reward rate was calculated as the average outcome earned divided by average outcome available, which was determined by the randomly drifting outcome distributions generated for each subject. **c.** Reaction time (milliseconds, ms) boxplots by study and task variant. Subjects made first-stage decisions quicker for the protection task variants compared to all other task variants. Far right legend indicates task variants across all studies: Study 1 = Reward and Protection 1, Study 2 = Direct Reward and Protection 2, Study 3 = Punishment and Protection 3. (TIF)

**S2 Fig. Stakes and model-based control.** We assessed whether use of model-based control was affected by stakes by testing whether stakes moderated stay behavior in mixed model analyses. **(a)** Stakes did not significantly interact with either the model-based or model-free component across tasks in Study 1: MB Estimate = .0001, SE = .009, z = .02, p = .988, 95% CI [-.02, .02], $\tau 00$ = .48, R2 = .15; MF Estimate = -.02, SE = .02, z = .89, p = .375, 95% CI [-.02, .05], $\tau 00$ = .49, R2 = .17. **(b)** Task duration interacted with stakes and previous outcome, such that there was no effect of stakes at the start of the task but high-stakes trials had an increase in likelihood of stay behavior at the end of the task: Estimate = -.001, $SE$ = .0003, $z$ = -2.10, $p$ = .036, 95% CI [-.001, -.00004], $\tau_{00}$ = .48, $R^2$ = .16. **(c)** Study 2, which increased trials to 200 non-practice (compared with 100 non-practice in Study 1), which revealed a stakes effect that interacted with the model-based component: Estimate = .01, $SE$ = .006, $z$ = 2.20, $p$ = .028, 95% CI [.002, .03], $\tau_{00}$ = .51, $R^2$ = .16. This effect was driven by the direct reward variant: direct reward Estimate = .02, $SE$ = .009, $z$ = 2.68, $p$ = .007, 95% CI [.006, .04], $\tau_{00}$ = .59, $R^2$ = .17; protection Estimate = .004, $SE$ = .009, $z$ = .37, $p$ = .711, 95% CI [-.02, .02], $\tau_{00}$ = .66, $R^2$ = .21. No significant interaction was present for the model-free component: Estimate = -.007, $SE$ = .01, $z$ = -.53, $p$ = .595, 95% CI [-.03, .02], $\tau_{00}$ = .51, $R^2$ = .17. **(d)** In Study 3, the stakes effect was not significant

with respect to either model-based or model-free component: MB Estimate = .006, $SE$ = .007, $z$ = .88, $p$ = .381, 95% CI [-.01, .02], $\tau_{00}$ = .46, $R^2$ = .17; MF Estimate = .02, $SE$ = .01, $z$ = 1.87, $p$ = .062, 95% CI [-.001, .05], $\tau_{00}$ = .46, $R^2$ = .18.
(TIF)

**S3 Fig. Study 2 stakes effect by task.** Diverging from Study 1, Study 2 revealed a stakes effect such that model-based weighting ($\omega$) differed between tasks for both high and low stakes, with the protection variant demonstrating more model-based control for both stakes: high stakes $w_{protection}$ = .70(.21), $w_{direct.reward}$ = .55(.23), $t(199)$ = 7.40, $p < .001$, 95% CI [.11, .19], low stakes $w_{protection}$ = .77(.19), $w_{direct.reward}$ = .55(.23), $t(199)$ = 11.82, $p < .001$, 95% CI [.18, .26].
(TIF)

**S4 Fig. Reaction time by stakes and task variant.** RT for first-stage decisions only differed by stakes for Study 3, such that RTs were slower for high stakes: Estimate = 9.84, $SE$ = 4.75, $t$ = 2.07, $p$ = .039, 95% CI [.52, 19.16], $\sigma^2$ = 67.23, $\tau_{00}$ = 85.23, $R^2$ = .63.
(TIF)

**S5 Fig. Anxiety associations with metacognitive and predictive bias. (a)** Effects on Certainty of the interaction between anxiety (STAI) and task-variant by Study. **(b)** Effects on Outcome Estimates of the interaction between anxiety (STAI) and task-variant by Study.
(TIF)

**S1 Table. Additional model parameters.** Eligibility trace ($\lambda$), stickiness ($\pi$), and inverse-temperature ($\beta$) parameters by study for Model 3.
(XLSX)

**S2 Table. Model fit.** Watanabe-Akaike Information Criterion (WAIC) scores for each model by study.
(XLSX)

## Acknowledgments

We thank Alexandra Hummel for her help with task development.

## Author Contributions

**Conceptualization:** Sarah M. Tashjian, Dean Mobbs.

**Data curation:** Sarah M. Tashjian.

**Formal analysis:** Sarah M. Tashjian, Toby Wise.

**Methodology:** Sarah M. Tashjian, Toby Wise.

**Project administration:** Sarah M. Tashjian.

**Supervision:** Dean Mobbs.

**Visualization:** Sarah M. Tashjian.

**Writing – original draft:** Sarah M. Tashjian.

**Writing – review & editing:** Dean Mobbs.

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
