## [Decision Letter · Decision Letter 0]

4 Jul 2022

Dear Dr. Tashjian,

Thank you very much for submitting your manuscript "Model-based prioritization for acquiring protection" for consideration at PLOS Computational Biology.

As with all papers reviewed by the journal, your manuscript was reviewed by members of the editorial board and by several independent reviewers. In light of the reviews (below this email), we would like to invite the resubmission of a significantly-revised version that takes into account the reviewers' comments.

Although all reviewers are enthusiastic with the topic, two of them raised important issues that need to be addressed. We hope you can make efforts to convince the reviewers and improve the clarity of the paper.

We cannot make any decision about publication until we have seen the revised manuscript and your response to the reviewers' comments. Your revised manuscript is also likely to be sent to reviewers for further evaluation.

Sincerely,

Ming Bo Cai

Associate Editor

PLOS Computational Biology

Natalia Komarova

Deputy Editor

PLOS Computational Biology

Although all reviewers are enthusiastic with the topic, two of them raised important issues that need to be addressed. We hope you can make efforts to convince the reviewers and improve the clarity of the paper.

Reviewer's Responses to Questions

**Comments to the Authors:**

Reviewer #1: I have previously reviewed this manuscript twice at another journal. The authors addressed and incorporated all my points. I have read through the manuscript again and I can't find any other points for the authors to address. I think this manuscript will make an excellent contribution to the field and will be of interest to a broad audience.

Reviewer #2: Tashjian et al. address the question of model-based (as opposed to model-free) control in the context of acquiring protection, as well as in (asymmetric) relation to reward acquisition (i.e., same valence but different context) and harm avoidance (i.e., same context but different valance). To this aim, a well-established two-step task was used across 3 studies using online samples, and the extent of model-based control for protection was found to be consistently higher than reward and punishment tasks. The task/model behavior was further assessed with metacognition and trait anxiety.

This work is very timely and interesting. First, it focused on protection, which was rarely investigated in the literature. Second, it draws direct comparisons regarding the asymmetric relationship between protection and reward/punishment. Overall, the analyses are carefully performed and mostly in support of the key conclusions. The paper is also very well-written. I do have some questions (see below) regarding some of the analyses, and hopefully they help improve the paper.

Major points:

(1) Conceptually, the 2x2 distinction between context and valance (Fig 1) is very nice and informative. But to make it really complete, a monetary loss condition should be considered. I am not sure how easily this can be done online, but if not feasible, a comprehensive discussion might be required.

(2) A few points regarding “Stake”

(2a) It seems that the manipulation of stake was not explicitly introduced in the background, and to be honest, I had to go a bit back and forth to figure out what it meant. So I’d appreciate if this manipulation could be made more evident in either the Intro or the beginning of the Results section.

(2b) Stake might not only affect the degree of model-based control, but also the exploration-exploitation trade-off. If I got it right, the stake manipulation (x1 vs x5) was presented in a pseudorandomized order. So if a participant had learned the task structure well, she might just want to perform well, irrespective of whether a 1 or 5 will be multiplied. On top of that, if x5 is presented, she may want to maximize the protection given the learned knowledge (ie exploitation), yet if x1 is presented, i.e., the “risk” is low, she may explore the alternative to find out if the reward schedule had changed (exploration). That said, a candidate model that differs in the softmax inverse-temperature shall be considered.

(3) I find the way to present modeling results first then followed by LME results a bit counter-intuitive. For me, the LME results are model-free/model-agnostic because it does not yet rely on the modeling results; rather, the main effect vs interaction “infers” the MB and MF component. I would first show the LME data, then the modeling/parameter results. Also, the LME result cannot “[…] validate the computational modeling analyses […]” (page 6). Instead, the modeling analysis explains the LME findings. To truly validate the modeling results, the authors may consider examining the effect of positive and negative prediction errors (cf. Fig 4, Kool et al., 2017). This way, model-derived decision variables can be connected with the observed behavior, hence validating the modeling results.

(4) I wonder how omega (MB weight) and alpha (learning rate) are correlated? I am asking because instead of running separate correlations between omega/alpha and the reward rate, a linear regression is more proper: reward rate ~ omega + alpha. This way, the potential correlation between omega and alpha is implicitly considered in the regression model.

(5) A slightly more motivated description of the models is needed (at the beginning of Page 6). In all the results section, it states that Model 3(or 4) was the best, but it is hard for anyone to know what Model 3 actually is. And, although it seems that the model is well developed, one has to dig into some of the original papers to know the exact model formulae. So a more detailed modeling section would be really beneficial in the Methods section; this is also to make the paper more appropriate for Plos CB. Last, since the authors used Stan for their model fitting, I highly encourage the authors also share their model code (so far I only task code and data is shared on osf; it is worth also sharing the analysis code). This practice is also in line with the open science policy of Plos CB.

(6) How initial values of the model was constructed, when the first 20 trials (Study 1 case) were not used in the analysis? It is likely that after 20 trials, participants have already learned at least something of the second-stage values.

(7) I am unsure about whether model parameters can be compared (Page 8) if the winning model is not the same. Having additional parameters (Model 4) may take out some variance that omega and alpha could have explained (Model 3) – essentially shifting the marginal distribution from the joint parameter space.

(8) Was working memory also measured in addition to metacognition and anxiety? It has been argued that working memory is associated with performance in the two-step task (eg. Collins et al., 2020).

Minor points:

- At least some of the main statistics should be reported when describing the LME results (on staying probability).

- Page 13, “[…] was performed using weakly informative prior distributions” I guess some transformation was also used (for example, omega, alpha), right? The authors may want to consider following the hBayesDM package paper (Ahn et al. 2017) for model detailed model description.

- I am a bit concerned by the learning rate results (Fig2b) – they seem widespread, and many of them are close to 0 or 1. I imagine this would hardly be the case if a hierarchical model was used. So related to my point #5, it would be beneficial to share the Stan code.

- I am keen to see the actual model comparison results (i.e. WAIC scores) in the results section or SI.

References:

Ahn, W. Y., Haines, N., & Zhang, L. (2017). Revealing neurocomputational mechanisms of reinforcement learning and decision-making with the hBayesDM package. Computational Psychiatry (Cambridge, Mass.), 1, 24.

Collins, A. G., & Cockburn, J. (2020). Beyond dichotomies in reinforcement learning. Nature Reviews Neuroscience, 21(10), 576-586.

Reviewer #3: The authors apply the computational framework of reinforcement learning – which has had enormous success in characterizing reward-based learning & decision-making across many contexts – to a novel context: acquiring “protection”, i.e. things that will reduce or prevent future losses. Protection acquisition has been studied as a maladaptive trait in clinical psychology, but not so much as an adaptive computational mechanism under the RL umbrella. In applying RL to protection acquisition, the authors find what to me was a surprising result: People are substantially more model-based when acquiring protection as opposed to just seeking rewards or avoiding punishments. The authors convincingly demonstrate this fact in three pre-registered experiments, and relate this behavior to metacognitive accuracy and anxiety.

I think this is a cool paper, and should be published. Extending RL to the broader range of learning/decision-making contexts that humans experience in their lives – such as protection acquisition – is important & timely, and this paper executes on it well. I applaud their transparency (e.g. with pre-registration) and was convinced of their veracity of their results.

I found myself tripped up, however, on some conceptual confusions that I’d love to see addressed. I’m also not totally convinced about the authors’ explanation for their results, and am worried about deflationary alternatives. As such, I recommend a substantial R&R.

Major points:

- Subjects could have just treated your protection task variant as a reward task variant – the protection task (if I understand correctly) is formally equivalent to the direct reward task, except with a negative constant added to the reward function (i.e. you just take whatever reward you got and subtract nine). Am I missing something, or is that right? If I am missing something, then you need to explain the task way more clearly. (You could say it’s different because “shields” are not a primary reward – but then, neither are “fairy coins”, so there’s really no difference.)

Assuming I’m not missing something, I think this raises two related concerns. First, is there really reason to think that protection acquisition is conceptually different from the other types of reward learning? I couldn’t quite figure out your argument for this in the introduction. Like, you hinged a lot on the difference between reward learning & protection acquisition being that one is in an appetitive context and the other is in an aversive context. Maybe I’m not enough in the appetitive vs aversive literature, but that felt weak to me. Like, in real life, aren’t you pretty much encountering both rewards and punishments all the time? (When you build a fence to keep out dangerous animals, you might have built it with your friend and had a good social experience; or taken a break to drink some ice-cold lemonade; or felt a gentle breeze on your face. Is this an aversive context because you’re thinking about how to keep out dangerous animals? Or an appetitive context because there’s lots of rewards? Given that people treat secondary rewards like money as rewarding/appetitive, why would the protection itself not be treated like an appetitive stimulus? Since the aversive thing is typically not going to happen until far in the future, protection acquisition seems more appetitive than aversive to me!) And then, in your actual task, the only thing that differs between the two is a constant in the reward function. Is that really enough to call it a fundamentally different context? Clearly that manipulation did actually change people’s behavior (although see the next paragraph) – am I just missing how much people really treat “positive reward function context” vs “negative reward function context” as fundamentally distinct? I’m totally open to being convinced of that, but I wanted an argument for it more explicitly (addressing these issues).

- The second, related concern is that there’s a boring explanation for why people are more model-based in the protection variant. My worry was that it’s just something like: The shields variant is just weirder or less natural for people, and puts them on “high alert” in a way. Like, getting a thing which then prevents another thing feels like more cognitive steps to me somehow than the other variants (even the one with the sacks? I didn’t really get that one anyway – how did the sacks differ from coins?).

You might come back and say: That’s exactly our hypothesis! That the “valence-context asymmetry” inherent in protection acquisition necessitates more model-based control. But this argument doesn’t sit well with me. First, the way you put it in the text is that valence-context asymmetries may require more “flexible action policies”, and you hint that this has something to do with the fact that non-protection cases have more “predictable environments”. But the tasks are formally equivalent except for a constant in the reward function. Protection tasks don’t seem any more unpredictable to me; there’s no actual, formal need for more flexibility in the protection variant than the other variants. (Another way to put this is that the model-based advantage – i.e. how much more reward a model-based algorithm got on average vs a model-free algorithm – would not be higher for the protection variant vs. the other variants of your task.)

Moreover, if you think I’m right that the increase in model-basedness is due to some kind of “high alert / weirdness” thing, then that’s not at all specific to protection. Anything I did to make the task weirder or less natural would cause it. For instance, imagine I told people that a random varying amount (positive or negative) would be added to their bonus each trial. That would not induce a consistent valence-context asymmetry in the same way – would it still make people equally model-based? I kinda think it would. Or imagine that I did a weird variant where fewer coins meant they received more bonus money at the end.. I'd make the same prediction there.

You might say, “Fine, the cause – inducing high-alert-ness – is extremely unspecific to protection, but as a matter of fact it *does* apply to protection cases, so it *will* actually make people more model-based in those cases”. That’s fine – but then I think the framing needs to be different. If that’s what you think is going on, you can’t frame it as something remotely unique to protection cases, and you’d have to really emphasize that this just happens to be a feature of protection that is making people more model-based. (Also, if this is what’s going on, it would make me worry about generalizability – like, do protection cases actually put people on high alert in real life?)

In contrast, if you disagree with me that my “high alert / weirdness” hypothesis is what’s explaining the difference between conditions, then I think you need to:

(a) make an argument for why valence-context asymmetry (i.e. adding a constant to the reward function) actually requires more flexibility / model-basedness, or identify a different reason why a valence-context asymmetry would engender more model-based-ness;

(b) give some evidence (or reason to think) that “high alert / weirdness” is not explaining the effect.

If you disagree with my “high alert / weirdness” hypothesis, the strongest thing I think you could do for the paper would be to propose a compelling alternative and run another experiment that adjudicates between them. But I’m not at all requiring that for revision, and I think it’s totally possible that you could convince me by doing some analyses of the existing data you have. For example: Could the stakes conditions help inform these questions somehow? Like, I know higher stakes should also put people on higher alert.. Did you find a stakes effect in Studies 2-3? I couldn’t really tell from your description of the results.

Another analysis you could run that would be informative for this question is to look at whether people stay consistently more model-based in the protection variant throughout the experiment. If it’s really some kind of “oh this is weird, I should be more careful” thing going on, then I’d predict you should only really find the increased model-based-ness in the first half of the experiment and not the second.

(You might turn to the fact that people are no slower – in fact, they’re faster? – in the protection variant, as evidence that people don’t find the task weirder. But I’m not really convinced by that. It fits with the “high alert” hypothesis in my head.)

Just to add one more thing to this train of thought: Even if there’s no meaningful formal difference between protection acquisition cases and other types of reward learning, maybe there’s still a *psychological* difference? Like, people categorize it differently in their heads? Is that what you think is going on? For instance, I’d be really curious to see a version of your protection variant where, instead of framing it as protection, you frame it as just subtracting a constant amount from your reward every time. Do you predict that people would still be more model-based there? If yes, that would be a very strong test of your hypothesis. If no, then is it really about a valence-context asymmetry, or is it something else (and what is it)?

Anyway, there’s a ton of thoughts in there – as you can see, I found myself a bit jumbled on these questions. If you can find some way to convincingly clarify these issues, I would be enthusiastic about this paper being published :).

Some smaller things:

- I was very confused by the task at first read (and still am a bit confused, even after digging into the methods section). I couldn’t tell whether it was always the same amount of flames on each trial (assuming the same stakes condition), or whether that varied randomly across trials. I couldn’t tell what the sacks did. Don’t make the reader dig into methods section (or, God forbid, the Supplement) to understand these things.. I think you should do some work to make the task description way clearer.

- I had a couple concerns about the model comparison method. I’d never heard of the WAIC before. At first I assumed it was just another criterion like AIC, BIC, or DIC which tries to correct for overfitting with the raw number of parameters (which is a really bad way to do it). But then I looked into it and realized it’s more complex than that in a way I didn’t exactly understand? Anyway, I think it would be really helpful to explain & justify the use of the WAIC here, for folks like me who don’t know it :).

Also, I was always taught that the best ways to do model comparison were to either use the random effects method from Stephan et al (i.e. estimate the model evidence using the Hessian matrix, treat model parameters as random effects across subjects, and compute exceedance probabilities; e.g. https://www.sciencedirect.com/science/article/abs/pii/S1053811909002638, http://www.cns.nyu.edu/~daw/d10.pdf, https://github.com/sjgershm/mfit), or to just do normal cross-validation. Is there a reason to do an asymptotic approximation like WAIC here, instead of one of those methods?

- In the fourth paragraph, you write: “In prior work, reward acquisition and punishment avoidance elicit similar weighting of model-based control.” But then later, you write: “… in line with prior work showing aversive contexts decrease model-free contributions to reward learning.” Are those contradictory, or am I missing something?

- For the metacognitive analysis: Don’t you need to do some really fancy stuff to correctly analyze metacognitive data? E.g. see Fleming & Lau (2014), “How to measure metacognition”, https://www.frontiersin.org/articles/10.3389/fnhum.2014.00443/full. They say you have to estimate an ROC curve, etc etc. I know the metacognitive stuff is not the main point of your analysis, so this may not be worth it, but it’s worth considering (and, if you don’t do it, justifying why you don’t do it).

- I didn’t understand the anxiety analysis. Are you saying you found a three-way interaction between study1/2 vs 3, model-based-ness, and STAI score? How should we interpret that? Are the main effects significant within study? When I first read the result, I thought you meant that high-anxiety people had a monotonic ordering of model-basedness in reward seeking > protection acquisition > punishment avoidance.. Is that right, or is it some kind of crossover interaction that I’m not understanding? It took me forever to figure out how that result mapped onto the graph in Figure 4. I think you should graph that by study separately (as you do in the other figures); I initially just read the graph as random noise differing b/w studies. I still don’t really know how to interpret that graph. Also, the primary analysis I was expecting was a first-order correlation b/w anxiety and model-based-ness (or model-free-ness) in protection acquisition, ignoring the other conditions. Do you find that? If not, what are the implications?

p.s. I want to apologize for my review being so late. Sorry for holding up the review process!

**Have the authors made all data and (if applicable) computational code underlying the findings in their manuscript fully available?**

Reviewer #1: Yes

Reviewer #2: **No: **analysis code and model code is not yet shared.

Reviewer #3: Yes

PLOS authors have the option to publish the peer review history of their article (what does this mean?). If published, this will include your full peer review and any attached files.

Reviewer #1: No

Reviewer #2: No

Reviewer #3: **Yes: **Adam Morris
---

## [Decision Letter · Decision Letter 1]

30 Nov 2022

Dear Dr. Tashjian,

Thank you very much for submitting your manuscript "Model-based prioritization for acquiring protection" for consideration at PLOS Computational Biology. As with all papers reviewed by the journal, your manuscript was reviewed by members of the editorial board and by several independent reviewers. The reviewers appreciated the attention to an important topic. Based on the reviews, we are likely to accept this manuscript for publication, providing that you modify the manuscript according to the review recommendations.

Please consider the additional comments from Reviewer 2. The editors will check the revision without sending out for another review.

Sincerely,

Ming Bo Cai

Academic Editor

PLOS Computational Biology

Natalia Komarova

Section Editor

PLOS Computational Biology

Please consider the additional comments from Reviewer 2. The editors will check the revision without sending out for another review.

Reviewer's Responses to Questions

**Comments to the Authors:**

Reviewer #2: The authors have done a comprehensive revision that considerably addressed a number of concerns raised in my initial comments. The conclusions are now better supported by the results.

Here I have two additional points, more for clarification.

(1) In R2.4 “[…] On top of that, if x5 is presented, she may want to maximize the protection given the learned knowledge (ie exploitation), yet if x1 is presented, i.e., the “risk” is low, she may explore the alternative to find out if the reward schedule had changed (exploration)”, the authors did not directly answer this question. Testing inverse-temperature resulting non-identifiability could indeed be an issue, but this does not mean the stake did not affect the potential exploration-exploitation trade-off – something that should be at least briefly discussed.

(2) In R2.5, I slightly disagree that the authors chose to follow the original way of presentation (cf. Kool et al 2017). Speculatively, even when the original authors are using the same paradigm again, they may also revise the way of presenting results.

Here, I respect the authors’ decision. But it helps to explicitly mention that it is intended to follow Kool et al 2017 closely, though the way of presenting is somewhat counterintuitive.

Reviewer #3: The authors have addressed all my questions/concerns. I appreciate their careful and nuanced responses. I think this is a fantastic paper, and I look forward to citing it.

**Have the authors made all data and (if applicable) computational code underlying the findings in their manuscript fully available?**

Reviewer #2: Yes

Reviewer #3: Yes

PLOS authors have the option to publish the peer review history of their article (what does this mean?). If published, this will include your full peer review and any attached files.

Reviewer #2: No

Reviewer #3: **Yes: **Adam Morris

Figure Files:

Data Requirements:

Reproducibility:

References:

---

## [Editor Report · Decision Letter 2]

9 Dec 2022

Dear Dr. Tashjian,

We are pleased to inform you that your manuscript 'Model-based prioritization for acquiring protection' has been provisionally accepted for publication in PLOS Computational Biology.

Best regards,

Ming Bo Cai

Academic Editor

PLOS Computational Biology

Natalia Komarova

Section Editor

PLOS Computational Biology

---

## [Editor Report · Acceptance letter]

14 Dec 2022

PCOMPBIOL-D-22-00468R2 

Model-based prioritization for acquiring protection

Dear Dr Tashjian,

I am pleased to inform you that your manuscript has been formally accepted for publication in PLOS Computational Biology. Your manuscript is now with our production department and you will be notified of the publication date in due course.

With kind regards,

Zsofia Freund
